# Ketonuria Is Associated with Changes to the Abundance of *Roseburia* in the Gut Microbiota of Overweight and Obese Women at 16 Weeks Gestation: A Cross-Sectional Observational Study

**DOI:** 10.3390/nu11081836

**Published:** 2019-08-08

**Authors:** Helen Robinson, Helen Barrett, Luisa Gomez-Arango, H. David McIntyre, Leonie Callaway, Marloes Dekker Nitert

**Affiliations:** 1Department of Obstetric Medicine, Royal Brisbane and Women’s Hospital, Butterfield St, Herston, QLD 4029, Australia; 2Mater Research Institute, The University of Queensland, Brisbane, QLD 4101, Australia; 3School of Chemistry and Molecular Biosciences, The University of Queensland, St Lucia, QLD 4072, Australia; 4Women’s and Newborn Services, Royal Brisbane and Women’s Hospital, Butterfield St, Herston, QLD 4029, Australia

**Keywords:** microbiome, pregnancy, obesity, ketonuria, *Roseburia*

## Abstract

The gut microbiome in pregnancy has been associated with various maternal metabolic and hormonal markers involved in glucose metabolism. Maternal ketones are of particular interest due to the rise in popularity of low-carbohydrate diets. We assessed for differences in the composition of the gut microbiota in pregnant women with and without ketonuria at 16 weeks gestation. Fecal samples were obtained from 11 women with fasting ketonuria and 11 matched controls. The samples were analyzed to assess for differences in gut microbiota composition by 16S rRNA sequencing. Supervised hierarchical clustering analysis showed significantly different beta-diversity between women with and without ketonuria, but no difference in the alpha-diversity. Group comparisons and network analysis showed that ketonuria was associated with an increased abundance of the butyrate-producing genus *Roseburia.* The bacteria that contributed the most to the differences in the composition of the gut microbiota included *Roseburia*, *Methanobrevibacter*, *Uncl. RF39,* and *Dialister* in women with ketonuria and *Eggerthella*, *Phascolarctobacterium*, *Butyricimonas,* and *Uncl. Coriobacteriaceae* in women without ketonuria. This study found that the genus *Roseburia* is more abundant in the gut microbiota of pregnant women with ketonuria. *Roseburia* is a butyrate producing bacterium and may increase serum ketone levels.

## 1. Introduction

Pregnancy is a time of metabolic and hormonal change. Ketogenesis is accelerated in pregnancy, particularly in the third trimester. Ketones are produced from the breakdown of lipids when the mother’s metabolic needs can no longer be met by glucose. The body produces three ketone bodies, beta-hydroxybutyrate, acetoacetate, and acetone. Beta-hydroxybutyrate and acetoacetate can be used as energy sources by the mother and the fetus and occur in a 1:1 ratio. Elevated maternal ketone levels have been associated with adverse fetal and childhood outcomes, particularly with regard to intelligence quotient (IQ), although results of these studies have been inconsistent [1,2,3,4].

The role of the gut microbiome in the metabolic changes of pregnancy has been an area of increasing interest. We have reported that the composition of the gut microbiome in pregnancy is associated with various metabolic and hormonal markers involved in glucose metabolism [5]. Whether the gut microbiome actually causes these metabolic changes is yet to be fully determined. However, mice colonized with the microbiome from women in the third trimester of pregnancy develop insulin resistance and increased adiposity, supporting the idea that the gut microbiome itself drives some of the metabolic changes observed in pregnancy [6].

Butyrate is a short-chain fatty acid (SCFA) that is produced by certain bacteria within the microbiome. Known butyrate-producing species include *Faecalibacterium prausnitzii*, *Roseburia* spp., and *Eubacterium rectale* [7]. Butyrate is the main fuel for energy production in colonocytes and studies of metabolism of human and rat colonocytes have shown that butyrate is metabolized to ketone bodies and carbon dioxide [8,9]. In keeping with this finding, mice colonized with *Roseburia* have higher serum levels of the ketone, beta-hydroxybutyrate [10].

There are no studies that have reported an association between the composition of the gut microbiome in pregnancy and maternal ketone levels. We hypothesized that gut microbiome composition is associated with maternal ketone levels and that butyrate-producing bacteria are more abundant in women with higher ketone levels.

## 2. Materials and Methods

Women enrolled in the Study of Probiotics IN Gestational diabetes (SPRING study) who supplied a stool sample and fasting urine sample at baseline (<16 weeks gestation) were included in this study. Women taking probiotics in pregnancy prior to 16 weeks gestation and sample collection were excluded from enrolment. This study was approved by the human research ethics committee of the Royal Brisbane and Women’s Hospital on the 16th January 2012 (HREC/11/QRBW/467) and The University of Queensland on the 25th January 2012 (201200080). All subjects gave their informed consent for inclusion before they participated in the study. The study was conducted in accordance with the Declaration of Helsinki. All women enrolled in the study were either overweight or obese, as defined by pre-pregnancy body mass index (BMI) > 25 kg/m^2^. Women collected a stool sample and fasting urine sample within a 24-h time period. Women fasted for between 9.5 and 12 h prior to collection of urine. The stool sample was kept in storage at −80 °C prior to fecal DNA isolation. Urine samples were immediately tested for the presence of ketones. Urine dipstick tests were performed using SIEMENS Multistix 10 SG reagent strips and measured levels of the ketone, acetoacetate. A ketone level of trace, small, moderate, and large corresponded to an acetoacetate level of 0.5 mmol/L, 1.5 mmol/L, 4 mmol/L, and >= 8 mmol/L, respectively. All women provided dietary information from the start of pregnancy by food frequency questionnaire (Cancer Council Victoria’s Dietary Questionnaire for Epidemiological Studies (Version (2)).

Women with any level of ketonuria present were matched with women with no ketonuria. The matching was performed on future GDM status, ethnicity, BMI and age. Fecal samples from each group were then analyzed to assess for differences in the gut microbiota between these two groups. Funding for this study was provided by the National Health and Medical Research Committee (NHMRC1028575) of Australia, the Royal Brisbane and Women’s Hospital Foundation, the Mater Foundation and the Australian Diabetes in Pregnancy Society. HB is funded by an NHMRC Early Career Fellowship. 

### 2.1. Fecal DNA Extraction

Stored stool samples were thawed at 4 °C before analysis. Aliquots of 250 mg of stool were removed from each sample for DNA extraction using the repeated bead beating and column (RBB + C) protocol. The aliquots were mixed with the RBB + C lysis buffer and sterile zirconia beads (0.1 and 0.5 mm diameter) and homogenized using a Tissue Lyser II (Qiagen, Chadstone VIC, Australia) for 3 min at 30 Hz. Samples underwent DNA purification using Qiagen AllPrep columns [5,11]. The quality and quantity of DNA was analyzed using the Nanodrop ND 1000 spectrophotometer (NanoDrop Technologies, Thermo Scientific, Scoresby, VIC, Australia) system.

### 2.2. Fecal Bacterial Identification

Bacteria within each sample were identified via 16S rRNA Sequencing. PCR amplification for the V6–V8 hypervariable regions of the bacterial 16S rRNA gene was performed using the 926F forward (50-TCG TCG GCA GCG TCA GAT GTG TAT AAG AGA CAG AAA CTY AAA KGA ATT GRC GG-30) and 1392R reverse (50-GTC TCG TGG GCT CGG AGA TGT GTA TAA GAG ACA GAC GGG CGG TGW GTR C-30) primers. Positive (*E. coli* JM109 DNA) and negative (deionized sterile water) controls were included in each PCR run. Nextera XT V2 index kit Sets A and B were used to barcode PCR products and the AMPure XP bead system (Illumina, San Diego, CA, USA) was used for purification. Barcoded DNA underwent quantification, normalization and pooling to develop sequencing libraries which were then sequenced on the Illumina MiSeq platform (Illumina, San Diego, CA, USA) at the Australian Centre for Ecogenomics at The University of Queensland. The Quantitative insights Into Microbial Ecology (QIIME) v1.9.1 analysis tool was used to join and de-multiplex forward and reverse sequences. Using the Greengenes reference database, the open reference operational taxonomic unit (OTU) picking method was used for taxonomic assignments with a pairwise identity threshold of 97%. Taxonomic units that were present in the negative controls were removed from the analysis along with OTUs with a relative abundance of <0.0001. Prior to downstream analysis, the OTU table was rarefied to 3000 sequences/sample with no samples removed in this step.

### 2.3. Statistical Analysis

Median and interquartile ranges (IQR) were used to present the data as bacterial abundance was not normally distributed. Non-parametric statistical methods were used and a *p* value of <0.05 was considered statistically significant. Sample profiles were analyzed via the online Calypso software tool [12] and results are presented at the genus level of taxonomic assignment. Chao1 and Shannon indices were used for comparison of alpha diversity (within sample diversity) and the Bray-Curtis dissimilarity index was used to assess beta diversity (between sample diversity). Network analysis was performed to identify positive and negative correlations between bacterial taxa for both patients with and without ketonuria. Genera associated with samples from women with and without ketonuria were identified using Spearman’s rho correlation coefficients with 1000-fold permutations. The strength of the color of the node reflects the significance of the association with either group and results are reported as significant if the false discovery rate (FDR) was <0.05. The size of the node reflects the abundance of the genus. Group comparisons were performed on genus level using the Wilcoxon Rank test, with no genera passing the statistical threshold for multiple testing, which is likely a reflection of the overall number of participants in this sub-study.

## 3. Results

Eleven women with ketonuria at 16-weeks gestation were matched with 11 women without ketonuria (Table 1). There were no differences in baseline BMI, maternal age, ethnicity, fasting blood glucose levels, future GDM status, or carbohydrate intake between the groups.

### Comparison of Gut Microbiome Composition

There was no difference in the alpha diversity at genus level (Chao1, Shannon index) between the two groups (see Figure 1A,B). There was also no difference between the two groups in beta diversity at genus level with unsupervised hierarchical clustering principle coordinates analysis (PCoA)(Bray-Curtis dissimilarity index) (See Figure 2A), but with supervised redundancy analysis (RDA) there was a significant difference (*P* < 0.0001; RDA analysis, see Figure 2B). Analysis of the variance in the beta-diversity displayed no significant difference between the groups (see Figure 2C).

Presence of urinary ketones was associated with an increased abundance of the butyrate-producing genus *Roseburia* in the network analysis. The brightness of the nodes is related to the level of significance of the association. *Roseburia* is the brightest of the nodes associated with ketonuria (see Figure 3); however, the butyrate-producer *Faecalibacterium* is also associated with ketonuria as is the acetate/propionate producer *Dialister*. In women who did not have ketonuria at 16 weeks, the abundance of *Adlercreutzia*, *Bifidobacterium*, *Dorea,* and *Collinsella* was higher (see Figure 3). Group comparisons revealed a statistically significantly higher abundance of *Roseburia* in women with ketonuria (see Figure 4A), and *Dialister* and *Faecalibacterium* abundance tended to be higher in the women with ketonuria (*P* = 0.066 and *P* = 0.076 respectively). In the women without ketonuria, *Adlercreutzia* abundance trended to be higher (*P* = 0.066), but that of *Bifidobacterium*, *Dorea,* and *Collinsella* was not significantly higher (*P* = 0.15; *P* = 0.14; and *P* = 0.17), respectively. 

The bacteria that contribute the most to the differences in the composition of the gut microbiota include *Roseburia*, *Methanobrevibacter*, *Uncl. RF39,* and *Dialister* in the women with ketonuria and *Eggerthella*, *Phascolarctobacterium*, *Butyricimonas,* and *Uncl. Coriobacteriaceae* in the women without ketonuria (see Figure 4B). Predicted bacterial functions that were increased in women with ketonuria included riboflavin metabolism, lipid biosynthesis, carbon fixation pathways in prokaryotes, zeatin biosynthesis, adipocytokine signaling pathway, biotin metabolism, folate biosynthesis, prenyltransferases, and peroxisome (See Appendix A). In women without ketonuria, bacterial functions ascorbate and aldarate metabolism; electron transfer carriers, phosphotransferase system PTS; aminobenzoate degradation, drug metabolism cytochrome P450; limonene and pinene degradation; chlorocyclohexane and chlorobenzene degradation; and styrene degradation were predicted to be more abundant. These results were all statistically significant (*p* < 0.05) on simple testing, but not after correction for multiple comparisons.

## 4. Discussion

This study shows that *Roseburia* is more abundant in the stool samples of women with fasting ketonuria at 16 weeks gestation. *Faecalibacterium* and *Dialister* species tended higher in the stool samples of women with ketonuria; however, this increased abundance did not reach statistical significance. Other studies have shown an association between the gut microbiota and hormonal and metabolic markers of glucose metabolism in pregnancy [5]. Our study expands on these findings and suggests that the microbiota may affect the capacity to produce ketone bodies during fasting. 

*Roseburia* and *Faecalibacterium* are both genera of obligate gram-positive anaerobic bacteria. These bacteria ferment carbohydrates in the colon to produce SCFAs, particularly butyrate. The presence of both *Roseburia* and *Faecalibacterium* species in the gut has been associated with human metabolism outside of pregnancy. In two large metagenome-wide association studies, concentrations of butyrate-producing bacteria, *Roseburia intestinalis* and *Faecalibacterium prausnitzii* were lower in patients with type 2 diabetes mellitus when compared with those with normal carbohydrate metabolism [13,14]. The abundance of *F. prausnitzii* has also been found to be significantly lower in obese patients when compared with lean subjects [15]. In contrast, *Roseburia* has been found to be significantly higher in patients with higher BMI [16]. *Roseburia* abundance is linked to overall carbohydrate intake with individuals with lower carbohydrate intake having lower *Roseburia* abundance [17]. In our study, carbohydrate intake was not significantly different between women with ketonuria (137.2 (95.7–171.2) g/day) and those without ketonuria (155.9 (122.0–170.1) g/day; *P* = 0.46), but this may be due to the small sample size. *Roseburia* spp. contain genes involved in riboflavin metabolism and folate biosynthesis indicating how the increases in the abundance in the predicted function analysis may be related to increased *Roseburia* abundance. In addition, *Dialister* spp. and *Faecalibacterium* spp. also express genes for riboflavin metabolism, folate biosynthesis, and biotin metabolism, which could further contribute to the predicted functional differences between the groups. 

Studies have found various links between butyrate-producing bacteria and serum ketone levels. Butyrate is the main energy source for colonocytes. In vitro studies of human and rat colonic cells show that the metabolism of butyrate involves oxidation and that part of the oxidized butyrate is converted to ketone bodies, acetoacetate, and beta-hydroxybutyrate [8,9]. A link between *Roseburia* and serum ketones has also been seen in mice. When mice were fed a diet with a high content of plant polysaccharides and colonized with a ‘core’ community of bacterial species including *Roseburia intestinalis*, they had higher levels of serum beta-hydroxybutyrate when compared with mice on the same diet, colonized with the same ‘core’ community without *Roseburia intestinalis* [10]. It is uncertain as to how these results can be extrapolated to humans, as the human gut microbiota is highly diverse and it is unclear if the presence of a single species would elicit a similarly large effect. Furthermore, dietary intake varies between individuals and across cultures. 

A different mechanism for the link between butyrate and ketone levels has been hypothesized. Intraperitoneal administration of butyrate into mice led to increased serum beta-hydroxybutyrate levels and fibroblast growth factor 21 (FGF21) levels [18]. The hormone FGF21 stimulates fatty acid metabolism in the liver leading to increased ketogenesis. The authors postulated that the increase in serum ketone levels was due to butyrate increasing FGF21 levels via induction of FGF21 gene expression in the liver. In humans, FGF21 is also expressed in the liver and is a downstream target of the transcription factor peroxisome proliferator-activated receptor alpha (PPARα). PPARα is a major regulator of lipid metabolism in the liver and is activated by both fasting and by consumption of ketogenic diets [19]. It is not known whether serum butyrate induces FGF21 gene expression in humans and what level of butyrate would be required to do so. In the current study, butyrate levels in the circulation were not measured and it is not clear if the increased abundance of *Roseburia* results in higher levels in the circulation. 

*Adlercreutzia* abundance trended to be higher in those without ketonuria. *Adlercreutzia* is an obligate anaerobic coccobacillus and has been linked to various inflammatory conditions including inflammatory bowel disease, primary sclerosing cholangitis and multiple sclerosis [20,21,22]. Studies linking *Adlercreutzia* to human metabolism are lacking. One study did find a reduced abundance of *Adlercreutzia* in patients with HIV who developed diabetes compared with those who did not develop diabetes [23]. *Eggerthella*, *Phascolarctobacterium*, *Butyricimonas,* and *Uncl. Coriobacteriaceae* were the bacteria that contributed most to the differences in the composition of the gut microbiota in women without ketonuria. In a metagenome-wide association study of GDM, *Phascolarctobacterium* was more abundant in women with GDM whereas *Eggerthella* was more abundant in healthy controls [24]. In the same study, when assessing metagenomic linkage groups, *Methanobrevibacter smithii* was enriched in healthy controls. In our study, *Methanobrevibacter* was one of the bacteria that contributed most to the differences in the composition of the gut microbiota in women with ketonuria. 

Maternal ketonuria has been associated with adverse fetal and childhood outcomes, particularly reduced childhood IQ [1,2]. However, these studies had disparate methodologies, with inconsistent results [3,4]. Ketone production occurs more rapidly in pregnancy, particularly in the third trimester [25]. It is felt to be due to increased maternal lipid metabolism and reduced glucose levels due to glucose being transported to the fetus for energy [26,27]. A maternal diet that is low in carbohydrate will also result in increased maternal ketone levels. The fetus utilizes ketones for energy and also as an important precursor for brain tissue [28]. It would therefore appear necessary for the fetus to be exposed to some level of ketones, however whether a high level of ketone exposure *per se* is harmful to the fetus is unclear. Our results suggest that ketone production in pregnancy may be more complex than a simple metabolic switch from glucose to lipid metabolism when maternal glucose supply is low. The gut microbiota may have a role to play via other metabolic pathways. 

Limitations of the study included that urine ketone levels were only measured at one time point in the first trimester. Women in the control group may have had ketonuria at other times from when their samples were collected. A strength of the study is that women fasted for at least 9.5 h prior to the collection of the urine and ketogenesis is more pronounced with increased duration of the fasting state. Our sample size was limited which may have reduced the power of the study, particularly in relation to any differences in dietary intake. Lastly, circulating SCFA levels were not measured, which could indicate whether circulating butyrate levels are different in women with and without ketonuria. 

## 5. Conclusions

*Roseburia* is more abundant in the microbiome of pregnant women with ketonuria. *Roseburia* is a butyrate-producing bacteria and studies have shown a link between both *Roseburia* and butyrate in the colon and elevated serum ketone levels. This study is further evidence of such a link and the first time that this link has been seen in pregnancy. Increased butyrate production by the gut microbiota may alter signaling in the host and thereby contribute to overall metabolic health in pregnancy. Larger studies of the microbiome in pregnant women with and without ketonuria at multiple time points in the pregnancy, with detailed dietary data needs to be done to further understand the relationship between *Roseburia* and maternal metabolism. 

## Figures and Tables

**Figure 1 nutrients-11-01836-f001:**
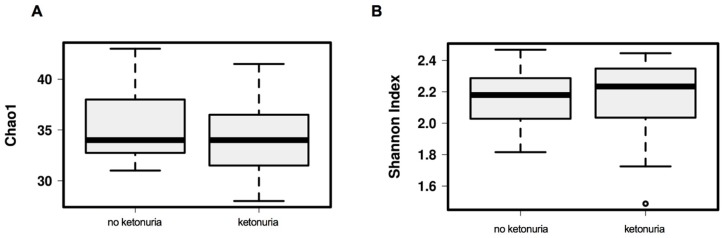
Alpha diversity of the gut microbiota at genus level between women with and without ketonuria. (**A**) Alpha diversity as assessed with the Chao1 index; (**B**) alpha diversity, as assessed with the Shannon index.

**Figure 2 nutrients-11-01836-f002:**
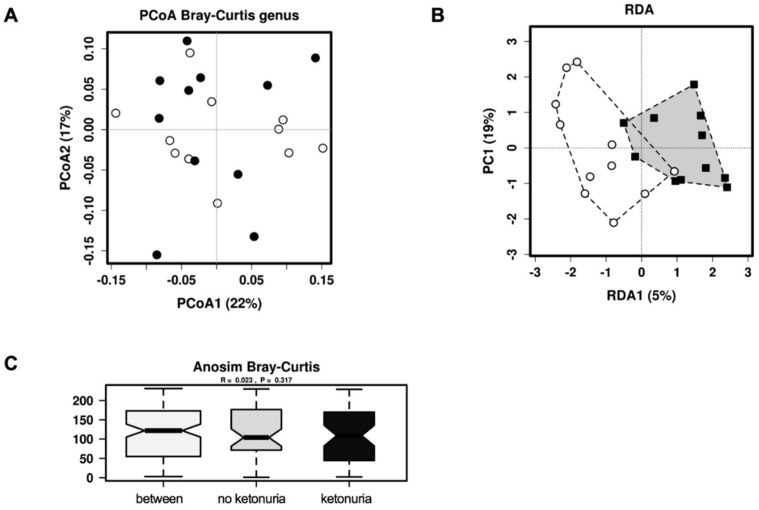
Beta diversity of the gut microbiota at genus level between women with and without ketonuria. (**A**) Unsupervised hierarchical clustering analysis by PCoA; (**B**) supervised clustering analysis by RDA and (**C**) variance analysis by Anosim analysis. Black circles/squares, ketonuria; white circles, no ketonuria.

**Figure 3 nutrients-11-01836-f003:**
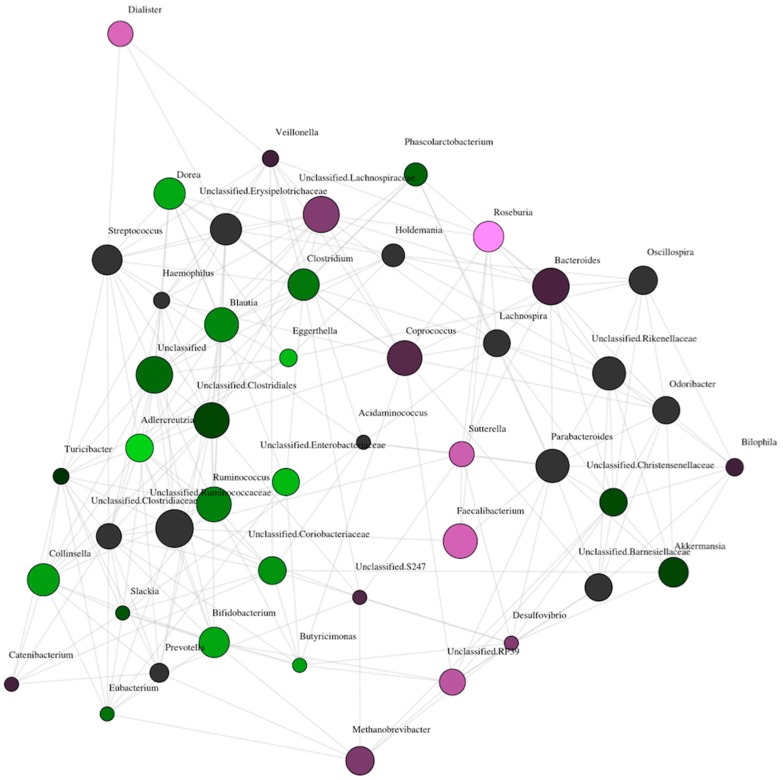
Network analysis of gut microbiota composition between women with and without ketonuria at genus level. Purple circles, ketonuria; green circles, no ketonuria; lines indicate positive correlations between the abundances of the bacteria. The size of the circle indicates the overall abundance of the genus and the brightness of the color indicates the degree to which the genus is associated with the group.

**Figure 4 nutrients-11-01836-f004:**
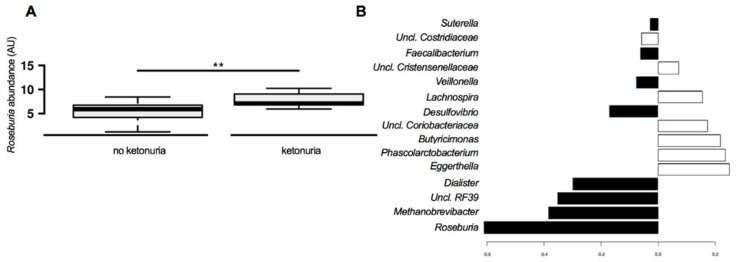
Bacterial genera that are specifically associated with the presence or absence of ketonuria. (**A**) Abundance of *Roseburia* between the groups. (**B**) sPLS-DA analysis of the contribution of bacteria genera to the differences between the gut microbiota between the groups. Black bars, ketonuria; white bars, no ketonuria. ** *p* < 0.01.

**Table 1 nutrients-11-01836-t001:** Participant characteristics.

	Ketonuria	No Ketonuria
N	11	11
Maternal age (years)	33 (29–38)	32 (29–33)
Gestation (weeks)	14 (14–15)	14 (14–15)
Maternal BMI (kg/m^2^)		
25–30 (%)	4 (36%)	4 (36%)
30–35 (%)	0 (0%)	0 (0%)
>35 (%)	7 (64%)	7 (64%)
Ethnicity		
Caucasian (%)	11 (100%)	11 (100%)
Fasting blood glucose (mmol/L)	4.4 (4.3–4.5)	4.2 (4.0–4.4)
Carbohydrate intake (g/day)	137.2 (95.7–171.2)	155.9 (122.0–170.1)
Later developed GDM (%)	1 (9%)	1 (9%)
Level of ketonuria		
Trace	8 (73%)	
Small	1 (9%)	
Moderate	1 (9%)	
Large	1 (9%)	

Data presented as median (IQR). BMI, body mass index; GDM, gestational diabetes mellitus; g, grams.

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
