# Peer review of "Ketonuria Is Associated with Changes to the Abundance of Roseburia in the Gut Microbiota of Overweight and Obese Women at 16 Weeks Gestation: A Cross-Sectional Observational Study"

_nutrients, 2019, doi:10.3390/nu11081836_

Round 1

Reviewer 1 Report

1. Since participants were selected for the Study of Probiotics IN Gestational diabetes, I suggest that the authors add information on whether the women were taking Probiotics prior to fasting in the Materials and methods section. There is animal data that suggests that within a day of ceasing probiotic treatments the microbiome reverts back to its pre-treatment composition. If the women were taking probiotics immediately before testing, then compliance to treatment should be noted.

2. "Significant" was misspelled on line 222.

3. You should add "Obesity" to your keyword list. This will make retrieval of your paper much easier for future systematic reviews.

I have no other comments and look forward to your future investigations. 

Author Response

Reviewer One:

Since participants were selected for the Study of Probiotics IN Gestational diabetes, I suggest that the authors add information on whether the women were taking Probiotics prior to fasting in the Materials and methods section. There is animal data that suggests that within a day of ceasing probiotic treatments the microbiome reverts back to its pre-treatment composition. If the women were taking probiotics immediately before testing, then compliance to treatment should be noted.

Response: Thank you, this is a valid point. However, women had their stool and urine samples collected at enrolment in the SRPING study and hence had not started taking probiotics or placebo as part of the study. Furthermore, any women that took probiotics by capsule prior to 16 weeks gestation were excluded from enrolment in the SPRING trial.  A sentence has been added to the manuscript to make this clearer.

“Women taking probiotics in pregnancy prior to 16 weeks gestation and sample collection were excluded from enrolment.”

"Significant" was misspelled on line 222.

Response: This has been changed

You should add "Obesity" to your keyword list. This will make retrieval of your paper much easier for future systematic reviews.

Response: Thank you, this has been added

I have no other comments and look forward to your future investigations. 

Response: Thank you for comments and positive feedback.

Reviewer 2 Report

This study conducted by Robinson et al. has shed light on the impact of gut microbiome on metarnal ketone levels. They concluded using patients in the third trimester by measuring microbiota composition by 16S rRNA sequencing that genus Roseburia, a butyrate-producing bacteria, is the reason why the ketone levels are much higher. The study is important to illustrate the relationship between the ketone levels and gut microbiota.

Author Response

Reviewer Two:

This study conducted by Robinson et al. has shed light on the impact of gut microbiome on metarnal ketone levels. They concluded using patients in the third trimester by measuring microbiota composition by 16S rRNA sequencing that genus Roseburia, a butyrate-producing bacteria, is the reason why the ketone levels are much higher. The study is important to illustrate the relationship between the ketone levels and gut microbiota.

Response: Thank you for your positive comments.

Reviewer 3 Report

Brief summary:

Helen Robinson et aldescribed the differences in the composition of the gut microbiota in pregnant women with and without ketonuria at 16 weeks of gestation using 11 women with fasting ketonuria and 11 matched controls. In their cohort study, ketonuria patient have higher abundance of Rosburia spp.

Broad comments 

This paper highlight an interesting observation that obese and overweight pregnant women with ketonuria have associated increase level of fecal Rosburia spp.The overall enthusiasm for this paper is decreased by the lack of possible mechanistic explanation. Would be interesting to have plasma sample to have circulating butyrate but also beta-hydroxybutyrate, a ketone body not measured by urine dipstick. As mentioned in discussion, ketone body increase following butyrate can come from colonic cell mainly as acetoacetate or through FGF21 liver activation and in normal liver condition with a BOH-butyrate to acetoacetate ratio of 1/1.

If the authors have collected blood sample at the time of urine ketone measurement and stool sampling, I would suggest to measure circulating Beta OH-butyrate/Acetoacetate ratio, butyrate as well as plasma FGF-21 to give more mechanistic insight to the discussion.

Specific comments 

Line 119: provide: I is missing

Line 120 facility: I is missing

Table 1: Please add carbohydrate consumption and mentioned it in results before discussion

Methods: How the carbohydrate intake is measured (questioner?) 

Results: I have difficulty to understand why the network analysis mentioned Faecalibacterium as well as Rosburia and Dialister as associated with ketonuria when the sPLS-DA analysis mentioned that the bacteria that contribute the most to the differences in the composition of the gut microbiota of patient with ketonuria include Roseburia, Methanobrevibacter, Uncl. RF39 and Dialister. In Figure 4,Faecalibacteriumis lower than Methanobrevibacter andUncl. RF39 but the association in network analysis (fig 3) seems higher.Please give some explanation if correct. 

Supp Figure 1: X axis is cropped 

Author Response

Reviewer Three:

Broad comments 

This paper highlight an interesting observation that obese and overweight pregnant women with ketonuria have associated increase level of fecal Rosburia spp.The overall enthusiasm for this paper is decreased by the lack of possible mechanistic explanation. Would be interesting to have plasma sample to have circulating butyrate but also beta-hydroxybutyrate, a ketone body not measured by urine dipstick. As mentioned in discussion, ketone body increase following butyrate can come from colonic cell mainly as acetoacetate or through FGF21 liver activation and in normal liver condition with a BOH-butyrate to acetoacetate ratio of 1/1.

If the authors have collected blood sample at the time of urine ketone measurement and stool sampling, I would suggest to measure circulating Beta OH-butyrate/Acetoacetate ratio, butyrate as well as plasma FGF-21 to give more mechanistic insight to the discussion.

Response: Thank you for your comments. We agree that measuring Beta OH-butyrate/Acetoacetate ratio and butyrate levels would be of value in this study as we have indicated in the limitations section; however, in this analysis we have elected to focus on ketonuria measured immediately. There is some evidence that acetoacetate is unstable and decreases even when samples are stored at -80 degrees C (Fritzsche et al. Clin Lab 2001; 47(7-8):399-403). In addition, ketonuria is far more common in pregnancy than ketonemia (Spanou et al. Hormones 2015; 14(4):644-650), with ketonemia only being present in women with severe ketonuria. Most of the women in this study only showed relatively mild ketonuria and it is therefore unlikely that the ketonemia measurements would result in meaningful results with this small set of samples.

With regards to FGF-21 levels, this is an interesting suggestion. We have previously shown that FGF-21 is variably expressed in the placenta in women with and without pregnancy complications but increased in women who have GDM even though circulating levels were not altered (Dekker Nitert et al. JCEM 2014; 99(4):E591-8 and Dekker Nitert et al. Reproductive Biology and Endocrinology 2015; 13:14). In these studies, we also measured circulating FGF21 levels and found these to be extremely variable in pregnant women. Given the high variability and the fact that circulating FGF21 could be both of hepatic and placental origin and the absence of hepatic or placental samples, we do not think that an analysis of FGF21 levels would result in a deeper understanding of the mechanisms by which the microbiota would affect hepatic FGF21 production.

Specific comments 

Line 119: provide: I is missing

Response: This has been corrected.

Line 120 facility: I is missing

Response: This has been corrected

Table 1: Please add carbohydrate consumption and mentioned it in results before discussion

Response: The carbohydrate intake data have been added to table 1. We also added a sentence to the results stating that there were no differences in baseline characteristics apart from the presence of ketonuria:

“There were no differences in baseline BMI, maternal age, ethnicity, fasting blood glucose levels, future GDM status or carbohydrate intake between the groups.”

Methods: How the carbohydrate intake is measured (questioner?) 

Response: All women completed a food frequency questionnaire detailing their dietary intake since the start of pregnancy. This has been clarified in the manuscript:

“All women provided dietary information from the start of pregnancy by food frequency questionnaire (Cancer Council Victoria’s Dietary Questionnaire for Epidemiological Studies (Version 2)).”

Results: I have difficulty to understand why the network analysis mentioned Faecalibacterium as well as Rosburia and Dialister as associated with ketonuria when the sPLS-DA analysis mentioned that the bacteria that contribute the most to the differences in the composition of the gut microbiota of patient with ketonuria include Roseburia, Methanobrevibacter, Uncl. RF39 and Dialister. In Figure 4,Faecalibacteriumis lower than Methanobrevibacter andUncl. RF39 but the association in network analysis (fig 3) seems higher.Please give some explanation if correct. 

Response:  Thank you. These analyses are based on different assumptions. The network analysis represents a visual representation of the abundance of bacterial genera that are positively correlated with each other and which are associated with one of the groups. The size of the node represents the overall abundance of that genus and the intensity shows the level of correlation with that group.

In contrast, in the sparse partial least square-discriminant (sPLS-DA) analysis, the aim is to determine the contribution of the bacteria that drive the variation in the microbiome composition between the groups. This method uses multivariate statistics and therefore a bacterium that is highly associated with ketonuria (e.g. Faecalibacterium) by itself, may contribute less to overall differences between the groups.

Supp Figure 1: X axis is cropped

Response: Thank you, this has been fixed.